# Eckol from *Ecklonia cava* Suppresses Immunoglobulin E-mediated Mast Cell Activation and Passive Cutaneous Anaphylaxis in Mice (Running Title: Anti-Allergic Activity of *Ecklonia cava*)

**DOI:** 10.3390/nu12051361

**Published:** 2020-05-09

**Authors:** Eui Jeong Han, Hyun-Soo Kim, K.K.A. Sanjeewa, K.H.I.N.M. Herath, You-Jin Jeon, Youngheun Jee, Jeongjun Lee, Taehee Kim, Sun-Yup Shim, Ginnae Ahn

**Affiliations:** 1Department of Food Technology and Nutrition, Chonnam National University, Yeosu 59626, Korea; iosu5772@naver.com; 2National Marine Biodiversity Institute of Korea, 75, Jangsan-ro 101 gil, Janghang-eup, Seocheon 33662, Korea; gustn783@naver.com; 3Department of Marine Life Science, Jeju National University, Jeju 63243, Korea; asanka.sanjeewa001@gmail.com (K.K.A.S.); youjin2014@gmail.com (Y.-J.J.); 4Department of Veterinary Medicine and Veterinary Medical Research Institute, Jeju National University, Jeju 63243, Korea; madushaniherath001@gmail.com (K.H.I.N.M.H.); yhjee@jejunu.ac.kr (Y.J.); 5Naturetech, 29-8, Yongjeong-gil, Chopyeong-myeon, Jincheon-gun, Chungbuk 27858, Korea; jjlee@naturetech.co.kr (J.L.); taeheek@naturetech.co.kr (T.K.); 6Fisheries Science Institute, Chonnam National University, Daehak-Ro, Yeosu 59626, Korea; 7Department of Marine Bio-Food Sciences, Chonnam National University, Yeosu 59626, Korea

**Keywords:** *Ecklonia cava*, Eckol, immunoglobulin E, bone marrow-derived cultured mast cells, passive cutaneous anaphylaxis

## Abstract

Eckol, a precursor compound belonging to the dibenzo-1,4-dioxin class of phlorotannins, is a phloroglucinol derivative that exerts various activities. In the present study, we investigated the antiallergic effects of eckol isolated from the marine brown algae, *Ecklonia cava* using immunoglobulin E (IgE)/bovine serum albumin (BSA)-stimulated mouse bone marrow-derived cultured mast cells (BMCMC) and a mouse model of anaphylaxis. Eckol inhibited IgE/BSA-induced BMCMC degranulation by reducing β-hexosaminidase release. A flow cytometric analysis revealed that eckol decreases FcεRI expression on cell surface and IgE binding to the FcεRI in BMCMC. Moreover, eckol suppressed the production of the cytokines, interleukin (IL)-4, IL-5, IL-6, and IL-13 and the chemokine, thymus activation-regulated chemokine (TARC) by downregulating, IκB-α degradation and NF-κB nuclear translocation. Furthermore, it attenuated the passive cutaneous anaphylactic reaction induced by IgE/BSA-stimulation in the ear of BALB/c mice. These results suggest that eckol is a potential therapeutic candidate for the prevention and treatment of allergic disorders.

## 1. Introduction

Marine algae are consumed by humans and used as medicinal sources owing to their biochemical properties and relative abundance [1,2,3]. Among marine seaweeds, brown algae (including *Ecklonia* species) have been investigated extensively as they contain phlorotannins, which possess various physiological properties. Furthermore, *Ecklonia cava*, brown algae in the family Lessoniaceae are characterized by abundant eckol-type phlorotannins [2,3,4]. Eckol is a precursor compound belonging to the dibenzo-1,4-dioxin class of phlorotannins and is a component of marine brown and red algae. This compound is a trimer of phloroglucinol and has been reported to possess antioxidant [5,6,7,8,9,10], anti-inflammatory [11,12], anticancer [13,14], antihepatic [10] activities, as well as inhibitory effects on acetylcholine esterase [15] and protective effects against skin damage [16]. However, the antiallergic effects of eckol in immunoglobulin E (IgE)/bovine serum albumin (BSA)-stimulated bone marrow-derived cultured mast cells (BMCMC) and a passive cutaneous anaphylaxis (PCA) animal model have not been investigated.

Allergies are immune responses to environmental factors, allergens such as house dust mites, pollen, food, insects, and medications, and allergic reactions are dependent of the toxicity of these factors. Various cells such as T and B lymphocytes, mast cells, eosinophils, basophiles, and keratinocytes play an important role in the induction, progression, and alleviation of allergic reactions [17]. In particular, mast cells and basophils act as effector cells, while granulocytes function in IgE-mediated allergic reactions. Mast cells and basophils have been implicated in the expression of a wide variety of biological responses including immediate hypersensitivity reactions, host defense against pathogens (e.g., parasites), and neoplasms, angiogenesis, tissue repairing, and immunologically nonspecific inflammatory signals [17]. Mast cells are hematopoietic cells that reside in virtually all vascularized tissues. These cells are potential sources of a wide variety of biologically active secreted products including various cytokines such as interleukin (IL)-1, IL-3, IL-4, IL-5, IL-6, IL-9, IL-11, IL-13, tumor necrosis factor (TNF)-α and interferon (IFN)-γ, as well as growth factors such as stem cell factor (SCF) and granulocyte-macrophage colony-stimulating factor (GM-CSF). These products regulate the development and activation of T cells, B cells, monocytes, macrophages and granulocytes such as basophils, eosinophils, and neutrophils [18,19,20,21]. In addition to mediators of inflammation, mast cells secrete cytokines that function in the crosslinking of FcεRI to allergen-specific IgE antibodies [22]. Allergen-specific IgE is produced by allergen-immunized B cells, which in turn tether to the IgE receptor on the surface of basophils and mast cells where they function as effector cells [23,24]. This crosslinking is the major stimulus for effector cells and it triggers degranulation, resulting in the secretion of various chemical mediators such as histamine, β-hexosaminidase, prostaglandins, leukotrienes, and cytokines to cause type I allergic reactions. Therefore, the regulation of mast cell activation is essential for the attenuation of type I allergic reactions.

In this study, we investigated the in vitro and in vivo antiallergic effects of eckol isolated from *E. cava* using IgE/BSA-stimulated BMCMC and a PCA animal model.

## 2. Materials and Methods

### 2.1. Material, Extraction, and Isolation

*E. cava* was collected from the Jeju Island, Korea. The dried powder of *E. cava* (10 kg) was extracted by stirring with MeOH (3 × 5 L) for 10 days. The extract (273 g) was suspended in water and partitioned with n-hexane (35.92 g), CH_2_Cl_2_ (20.49 g), EtOAc (24.87 g), and n-BuOH (106 g) in sequence. The EtOAc fraction (24.87 g), which exhibited the most potent antiallergic activity in IgE/BSA-stimulated BMCMC, was subjected to a silica gel flash chromatography elution with hexane/EtOAc/MeOH (gradient) to yield 10 subfractions (F1-F10). The F5 (378.39 mg) subfraction with the highest antiallergic activity was further purified by Sephadex LH-20 in the presence of MeOH only to isolate eckol (58.30 mg) and the structure of eckol was determined (Figure 1).

Eckol was obtained as light brown powder (lyophilized). C_18_H_12_O_9_; ^1^H NMR (DMSO-d6, 400 MHz) δ: 9.54 (1H, s, OH-9), 9.45 (1H, s, OH-4), 9.21 (2H, s, OH-2, 7), 9.16 (2H, s, OH-3′, 5′), 6.14 (1H, s, H-3), 5.96 (1H, d, J) 2.8 Hz, H-8), 5.80 (1H, d, J) 1.7 Hz, H-4′), 5.78 (1H, d, J) 2.8 Hz, H-6), 5.72 (2H, J) 1.7 Hz, H-2′, 6′). 13C NMR (DMSO-d6, 100 MHz): LREIMS m/z: 373.00 [M + H]^+^.

### 2.2. Animals

C57BL/6 and BALB/c mice were used for preparation of BMCMC and induction of PCA, respectively. C57BL/6 (6–12 weeks old) and BALB/c (8 weeks old) mice reared under specific pathogen-free conditions were obtained from Orient Bio (Gwangju, Korea). Mice were housed under conditions that included a constant temperature of 23 ± 1.5 °C, humidity of 55 ± 15%, and 12 h on 12 h off light cycle. The mice were provided food and tap water ad libitum. All procedures involving animals were approved by the Institutional Animal Care and Use Committee of Chonnam National University (No.CNU IACUC-YS-2019-5).

### 2.3. Cell Viability Assay

The cytotoxicity of eckol was measured by the 3-(4-5-dimethyl-2yl)-2-5-diphenyltetrazolium bromide (MTT) assay. BMCMC (2 × 10^4^ cells) were treated with different concentrations of eckol for 24 h and were then reacted with 15 μL of MTT stock solution (5 mg/mL) for 4 h. The formazan crystals within cells were dissolved in 150 μL of dimethylsulfoxide (DMSO), and the absorbance was measured at 540 nm using a microplate reader (SpectraMax^®^ M2/M2e, Molecular Devices, Sunnyvale, CA, USA).

### 2.4. Preparation and Stimulation of BMCMC

BMCMC were obtained from 6–12-weeks-old male C57BL/6 mice according to method described in our previous publication [25]. BMCMC were sensitized with 1 μg/mL of anti-dinitrophenyl (DNP)-IgE (Sigma-Aldrich, St, Louis, MO, USA) for 4 h and challenged with the synthetic allergen, 100 ng/mL of IgE-BSA (LSL Japan Inc., TYO, Japan) for 1 h.

### 2.5. Degranulation Assay

To assess the inhibitory effects of eckol on mast cell degranulation, β-hexosaminidase released in the medium was measured. BMCMC (2 × 10^5^ cells) were first incubated in the presence of eckol at various concentrations for 2 h and then sensitized using anti-DNP-IgE (1 μg/mL) for 4 h. The vehicle was 0.1% BSA-Tyrode’s buffer. The sensitized BMCMC were challenged with DNP-BSA for 1 h. The cells were lysed with 50 μL of 0.5% Triton 100 in Tyrode’s buffer, and the supernatant was treated with substrate buffer (4-p-nitrophenyl-N-acetyl-β-D-glucosaminide 1.3 mg/mL, 0.1 M sodium citrate, pH 4.5) for 1 h. The reaction was terminated by the addition of 0.2 M glycine (pH 10.7), and the absorbance was measured at 405 nm. The amount of β-hexosaminidase released from sensitized mast cells was calculated according to the previously described method [25].

### 2.6. Reverse Transcriptase-Polymerase Chain Reaction (RT-PCR) for Cytokine mRNA Levels

Cytokine mRNA levels were examined by RT-PCR. Total RNA was extracted using TRIzol reagent and cDNA was synthesized using a PrimeScript RT Reagent Kit (TaKaRa Bio Inc., Otsu, Japan). PCR was performed for 35 cycles using specific sense and antisense primers. Human glyceraldehyde-3-phosphate dehydrogenase (*GAPDH*) was used as an internal control. The primer sequences were as follows: for the *IL-1β*, sense 5′-TCC AGG ATG AGG ACA TGA GCA C-3′ and antisense 5′-GAA CGT CAC ACA CCA GCA GGT TA-3′; for the *IL-4*, sense 5-ACG GAG ATG GAT GTG CCA AAC-3′ and antisense 5′-AGC ACC TTG GAA GCC CTA CAG A-3′; for the *IL-5*, sense 5′-TCA GCT GTG TCT GGG CCA CT-3′ and antisense 5′-TTA TGA GTA GGG ACA GGA AGC CTC A- 3′; for the *IL-6*, sense 5′-CCA CTT CAC AAG TCG GAG GCT TA-3′ and antisense 5′-GCA AGT GCA TCA TCG TTG TTC ATA C-3′; for the *IL-13*, sense 5′-CAA TTG CAA TGC CAT CTA CAG GAC-3′ and antisense 5′-CGA AAC AGT TGC TTT GTG TAG CTG A-3′; for the *IFN-γ*, sense 5′-CGG CAC AGT CAT TGA AAG CCT A-3′ and antisense 5′-GGC ACC ACT AGT TGG TTG TCT TTG-3′; for the *TNF-α*, sense 5′-GGC ACC ACT AGT TGG TTG CTT TG-3 ’ and antisense 5′ GTT CTA TGG CCC AGA CCC TCA C 3′; for the *TARC*, sense 5′-TGA GGT CAC TTC AGA TGC TGC-3′ and antisense 5′-ACC AAT CTG ATG GCC TTC TTC-3′; for the *GAPDH*, sense 5′-CAT CCG TAA AGA CCT CTA GCC AAC-3′ and antisense 5′-ATG GAG CCA CCG ATC CAC A-3′. PCR conditions were set as 5 min of denaturation at 94 °C, 1 min of annealing at 55–60 °C, and a 20 min extension at 72 °C (TaKaRa PCR Thermal Cycler). PCR products were visualized by 1.5% agarose gel electrophoresis and ethidium bromide staining under UV transillumination (Vilber Lourmat, Marne la Uallee, France).

### 2.7. Enzyme-Linked Immunosorbent Assay (ELISA) for Cytokines Production

BMCMC (2 × 10^6^ cells) were incubated in the presence of different concentration of eckol for 1 h and sensitized using anti-DNP-IgE (1 μg/mL) for 4 h. The sensitized cells were then incubated with DNP-BSA for 24 h. The cytokine content in the supernatants of cell suspension were analyzed using ELISA kits according to the manufacturer’s instructions (Biolegend Inc., San Diego, CA, USA).

### 2.8. Flow Cytometric Analysis for Expression and Binding of IgE of FcεRI

The expression of FcεRI and interactions between IgE and FcεRI were analyzed by flow cytomery. The pretreated BMCMC were sensitized using anti-DNP-IgE (1 μg/mL) for 4 h. After 20 min, the cells were blocked using anti-CD16/CD32 monoclonal antibodies and incubated with primary antibodies. Anti-mouse IgE and FcεRI antibodies were used to detect IgE and FcεRI binding and FcεRI expression, respectively. The cells were stained with FITC-conjugated anti-mouse antibodies as secondary antibodies and then measured by flow cytometry (Beckman Coulter, Brea, CA, USA).

### 2.9. Western Blot Analysis

BMCMC (2 × 10^5^ cells) were incubated with eckol or a positive control (indomethacin) for 1 h and then sensitized with anti-DNP-IgE (1 μg/mL) for 4 h. Then, the cells were treated with DNP-BSA (100 ng/mL) for 30 min. Cytosolic and nuclear proteins were extracted from the cells using the NE-PER^®^ Nuclear and Cytoplasmic Extraction Kit (Thermo Scientific, Rockford, IL, USA). The protein concentrations in both cell lysates were measured using a BCA™ Protein Assay Kit (Thermo Scientific) according to the manufacturer’s instructions. Equal amounts of protein were separated by 10% SDS-PAGE and transferred to nitrocellulose membranes, followed by blocking with a buffer (20 mM Tris (pH 7.4), and 136 mM NaCl) containing 5% skim milk. The membranes were incubated primary antibodies (1:1000 dilution) and HRP-conjugated specific-secondary antibodies (1:3000 dilution). Bands were detected using an enhanced Super Signal West Femto Maximum Sensitivity Substrate Reagent Kit (Thermo Scientific, Rockford, IL, USA) and analyzed using Image J (US National Institutes of Health, Bethesda, MD, USA).

### 2.10. PCA Test

The in vivo effects of eckol were investigated using a PCA animal model. First, anti-DNP-IgE (500 ng) was intradermally injected into the dorsal skin of both ears of BALB/c mice. Eckol was applied on mice ear 2 h prior to anaphylaxis induction using a 1 mL syringe. The mice were then intravenously treated with 30 μL of DNP-BSA (10 mg) saline solution containing 4% Evans blue dye [26]. The control mice were treated with saline instead of eckol. After 30 min, the mice were euthanized by anesthetizing with isoflurane followed by cervical dislocation, and skin tissues were collected from the dorsal ear and were soaked in 1 mL of formamide overnight at 64 °C. The absorbance at 620 nm was measured to determine the amount of dye that was extravasated for the quantitative detection of vascular permeability.

### 2.11. Statistical Analysis

Comparisons were performed by one-way ANOVA and Duncan’s multiple range tests using SPSS Statistics v20. Values are reported as means ± standard error (S.E.), and *p* < 0.05 was considered significant.

## 3. Results

### 3.1. Effects of Eckol on β-Hexosaminidase Release in IgE/BSA-Stimulated BMCMC

To evaluate the cytotoxic effect of eckol on BMCMC, MTT assays were performed. As presented in Figure 2A, eckol did not exhibit any cytotoxic effect on BMCMC under any experimental concentrations. Subsequent experiments were performed using up to 100 μg/mL eckol. The degranulation of mast cells by the binding of IgE to cell surface FcεRI and IgE-specific antigens leads to the secretion of inflammatory mediators, such as histamine, β-hexosaminidase, leukotrienes, and prostaglandins, resulting in symptoms of allergic disorders [27,28]. β-hexosaminidase is a potent marker of mast cell degranulation. Stimulation with IgE/BSA resulted in greater β-hexosaminidase release than that in unstimulated and untreated cells, and β-hexosaminidase release was significantly reduced after pretreatment with eckol in a dose-dependent manner (Figure 2B).

### 3.2. Effects of Eckol on Cytokine Production in IgE/BSA-Stimulated BMCMC

Cytokines, chemokines, and growth factors play an important role in the activation, development, and migration of mast cells [21]. Cytokines are extracellular signaling proteins produced in response to stimuli such as antigen-specific IgE [29]. As determined by ELISA, eckol inhibited the production of Th2-type cytokines, such as *IL-4* (Figure 3A), *IL-5* (Figure 3B), and *IL-13* (Figure 3D) as well as proinflammatory cytokines, such as *IL-6* (Figure 3C).

### 3.3. Effects of Eckol on Cytokine mRNA Levels in IgE/BSA-Stimulated BMCMC

To confirm the eckol-mediated suppression of the gene expression of cytokines and a chemokine, RT-PCR analysis for the mRNA expression was conducted with specific primers using total cellular RNA prepared from eckol-pretreated and IgE/BSA-stimulated BMCMC.

As shown in Figure 4, the mRNA levels of cytokines such as *IL-1β*, *IL-4*, *IL-5*, *IL-6*, *IL-13*, *IFN-γ*, and *TNF-α*, increased dramatically by stimulation with IgE/BSA. However, mRNA expression were attenuated by treatment with eckol. In particular, eckol downregulated the mRNA expression levels of Th2-type cytokines such as *IL-4*, *IL-5*, and *IL-13*, although the 25 μg/mL eckol did not affect mRNA expression levels. Additionally, the mRNA expression levels of proinflammatory cytokines such as *IL-1β, IL-6,* and *TNF-α* were decreased by treatment with eckol from 50 μg/mL to 100 μg/mL in IgE/BSA-stimulated BMCMC. In addition, eckol dose-dependently decreased IgE/BSA-induced mRNA levels of Th1-type cytokines such as *IFN-γ*. Moreover, *TARC*, a chemokine with various chemical effects on immune cells [21], was suppressed by treatment with eckol (Figure 4B).

### 3.4. Effects of Eckol on NF-κB Activation in IgE/BSA-Stimulated BMCMC

NF-κB is found in the cytoplasm as an inactive complex bound to inhibitor kappa B (IκB) [30]. To determine if eckol inhibits NF-κB activation, we investigated NF-κB nuclear translocation and I κB-α degradation by western blotting. IgE/BSA stimulation induced an increase in the translocation of free NF-κB/p65 into the nucleus via IκB-α phosphorylation. Eckol inhibited the degradation of IκB-α within the cytosol (Figure 5A) and the translocation of the NF-κB/p65 subunit into the nucleus (Figure 5B) that were induced by IgE/BSA.

### 3.5. Effects of Eckol on FcεRI Expression and Binding of IgE to FcεRI.

Mast cells express the high affinity IgE receptor, FcεRI on their cell surface, and the cross-linking of FcεRI with allergen-specific IgE triggers an allergic response [27,28]. To examine the effects of eckol on cell surface FcεRI expression and the binding of IgE to FcεRI, eckol pretreated BMCMC were treated with anti-FcεRI antibodies and anti-DNP-IgE antibodies. Cells were assessed by indirect immunofluorescence and flow cytometry using anti-FcεRI and anti-DNP-IgE antibodies for cell surface FcεRI expression and binding of IgE to FcεRI, respectively. As shown in Figure 6A, cell surface FcεRI expression levels were 92.82%, 86.58%, 85.39%, and 78.46% for eckol concentrations of 0, 10, 25, 50, and 100 μg/mL, respectively. Additionally, the binding activity of DNP-specific IgE to cell surface FcεRI was 49.21%, 16.24%, 9.33%, and 2.52% for the same eckol concentration. The corresponding percentage in the negative control was 2.14% (Figure 6B).

### 3.6. Effects of Eckol on IgE/BSA-Induced PCA Reaction in Mice Model

To evaluate the in vivo antiallergic activity of eckol, its effect on IgE/BSA-induced PCA was examined using a BALB/c mice model. Evans blue dye enables the measurement of active, passive, and reverse PCA [26,31]. The ear subjected to anti-DNP-IgE stimulation should become dark blue after 10 min following dye administration, unlike the ear receiving PBS control treatment. As expected, and as shown in Figure 7, anti-DNP-IgE stimulation increased the apparent blue color in the ear. However, in mice supplemented with eckol, the intensity of the blue color decreased in a dose-dependent manner. Quantitative results for the dye extravasated from mouse skin tissues were in agreement with these qualitative observations.

## 4. Discussion

The incidence of allergic diseases is increasing rapidly owing to various environmental, genetic, and immunological factors. The use of macroalgae has attracted increasing attention in the fuel, cosmetics, pharmaceutical, and food industries [32,33]. Eckol isolated from *E. cava* is a phloroglucinol derivative known to possess various physiological properties such as antioxidant, anti-inflammatory, hepatoprotective, neuroprotective, antiobesity, antihypertensive, antibacterial, and antiviral activities [34]. In the present study, the antiallergic effects of eckol were examined for the first time using BMCMC and a PCA-induced animal model. To identify marine organisms that can be used for the treatment of allergic disorders, we isolated eckol from *E. cava* and examined its protective effects against allergic reactions using IgE/BSA-stimulated BMCMC and anaphylaxis-induced BALB/c mice. Mast cells were obtained from mouse bone marrow. These cells arise from pluripotent bone marrow progenitor cells that migrate into connective tissues containing numerous granules following bone marrow egression; within these tissues these cells develop into mature mast cells in response to stimulation by local growth factors. These cells then act as the first line of defense in the skin, airways, and gastrointestinal tract against external insults. Mature mast cells contain numerous granules that are rich in chemical mediators such as histamine, β-hexosaminidase, prostaglandins, and leukotrienes. The cytotoxicity of eckol as determined by the proliferation of BMCMC was examined to determine the appropriate eckol concentrations for use in all our subsequent analyses. Our results indicated that this compound exerted no toxic effects on cells at concentrations up to 100 μg/mL (Figure 2A). β-hexosaminidase is a potent degranulation factor in IgE-mediated allergic reactions. We determined that eckol inhibited β-hexosaminidase release in a dose-dependent-manner (Figure 2B).

Mast cells release a wide array of preformed and de novo synthesized proinflammatory and immunomodulatory mediators upon stimulation by IgE, allergens, or other stimuli. This immune response is mediated by various cytokines, such as IL-3, IL-6, IL-9, SCF, and GM-CSF which influence the development of granulocytes as well as cytokines, such as TNF-α, IFN-γ, IL-1β, IL-4, IL-5, IL-6, IL-9, IL-11, and IL-13, involved in the activation of macrophages and monocytes [21]. Our ELISA results showed that eckol inhibited the production of IL-4, IL-5, IL-6, and IL-13 (Figure 3). In particular, levels of IL-4 and IL-6 production were markedly reduced by the pretreatment with eckol in IgE/BSA-stimulated BMCMC at all concentrations examined. Additionally, an RT-PCR analysis indicated that eckol downregulated the gene expression of genes encoding Th2-type cytokines, such as *IL-4, IL-5,* and *IL-13*, Th1-type cytokines, such as *IFN-γ*, proinflammatory cytokines, such *IL-1β, IL-6,* and *TNF-α* (Figure 4A), and chemokine, such as *TARC* (Figure 4B). Interestingly, we found similar effects of eckol on the expression levels of *IL-5* and *IL-13* and levels of protein secretion, although the levels of *IL-4* and *IL-6* showed different patterns than those at the protein levels. In particular, IL-4 is the initial cytokine produced from mast cells after stimulation by IgE or other factors and activates the secondary cytokines, such as IL-5, IL-6, and IL-13 [35,36]. Although eckol weakly downregulated the gene expression of *IL-4*, it led to a marked reduction of its protein secretion in IgE/BSA-stimulated BMCMC. These results suggested that the capacity of eckol to reduce IL-4 secretion might affect the secretion of secondary cytokines, such as IL-5, IL-6, and IL-13. Moreover, the chemokine *TARC* was released by activated mast cells [22] and the gene expression of *TARC* was downregulated in response to eckol treatment (Figure 4B). Based on these results, we predict that eckol has an antiallergic effect by reducing the gene and/or protein expression levels of various cytokines and *TARC* as well as cell degranulation in IgE/BSA-stimulated BMCMC.

NF-κB is redox-sensitive transcription factor that functions in cytokine production and is predominately found in the cytoplasm as an inactive complex bound to IκB-α. Phosphorylated NF-κB is translocated to the nucleus, and phosphorylated IκB-α is degraded. The activation of NF-κB is mediated by the upregulation of adhesion molecules, chemokines, and inflammatory mediators. We performed western blotting to examine NF-κB translocation into the nucleus and IκB-α phosphorylation within the cytosol. We observed an inhibitory effect of eckol on IκB-α within the cytosol (Figure 5A) and NF-κB within the nucleus (Figure 5B). Therefore, in order to better understand the inhibitory effects of eckol on mast cell activation, further studies of the regulation of signaling factors, such as protein tyrosine kinase (Syk and Lyn), mitogen-activated protein kinases, and phospholipase Cγ1, are needed.

FcεRI is a high affinity IgE receptor that is expressed on the surface of mast cells and basophils. Crosslinking of FcεRI to allergen-specific IgE via the binding of multivalent allergens results in the activation of IgE-mediated allergic reactions [27,28]. Recently, extensive research has been focused on the identification of biologically active antiallergic compounds able to inhibit FcεRI expression and IgE binding to FcεRI. Among the bioactive phloroglucinol derivatives isolated from *E. cava*, fucodiphloroethol G, phlorofucofuroeckol A and eckol have been demonstrated to inhibit cell surface FcεRI expression and IgE binding to FcεRI [34]. Our results also demonstrated that eckol led to a slight reduction in FcεRI expression (Figure 6A) on the cell surface as well as a marked reduction of IgE expression (Figure 6B) bound to cell surface FcεRI. In addition, we found that eckol binds to the active site of IgE (PDB ID; 4GRG), based on a molecular docking analysis (data not shown). Pretreatment with eckol can result in strong binding to the active site of IgE, and the complex can effectively interrupt the binding of IgE to FcεRI. Finally, the BSA stimulation led to the lower mast cell activation due to reductions in both IgE and FcεRI expression reduced by eckol pretreatment. These results suggest that eckol suppressed mast cell activation, including the release of the degranulation marker, β-hexosaminidase, and the production of cytokines, reducing both cell surface FcεRI expression and IgE binding to FcεRI in IgE/BSA-stimulated BMCMC.

Finally, we examined the mechanism by which eckol suppresses IgE/BSA-mediated PCA in mice. PCA is characterized by an immediate skin reaction by a localized IgE-mediated allergic response in vivo, resulting in increased vascular leakage in the skin that can be assessed by an intravenous injection of Evans blue [33]. Eckol treatment reduced allergic inflammatory responses in the PCA-induced mice (Figure 7). Our results suggest that the inhibitory effect of eckol on both IgE and FcεRI expression effectively leads to the suppression of IgE-mediated allergic responses in IgE/BSA-mediated PCA mice. Further studies of the protective role of eckol in the regulation of FcεRI-mediated signaling factors are necessary to confirm its therapeutic applications in allergic disorders.

## 5. Conclusions

The results of this study demonstrated that eckol has antiallergic effects in IgE/BSA-stimulated BMCMC and PCA-induced mice. Our results indicated for the first time that eckol suppresses the activation of mast cells by inducing degranulation and cytokine production following exposure to IgE/BSA. The inhibitory effects of eckol were related to the downregulation of FcεRI expression and decreased IgE binding. Moreover, in our experiment on IgE/BSA-induced PCA mice, eckol exerted an antiallergic effect, which may be due to the downregulation of FcεRI expression and cytokine production. Taken together, our results suggest that *E. cava* containing eckol is a potentially effective functional food and pharmaceutical agent.

## Figures and Tables

**Figure 1 nutrients-12-01361-f001:**
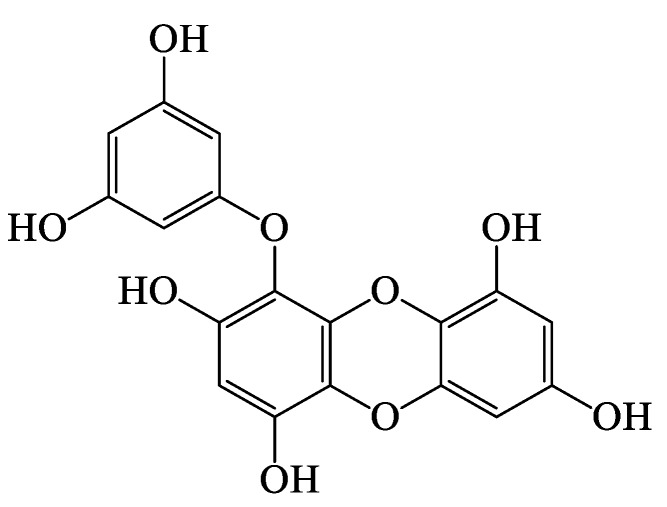
Chemical structure of eckol isolated from *E. cava.*

**Figure 2 nutrients-12-01361-f002:**
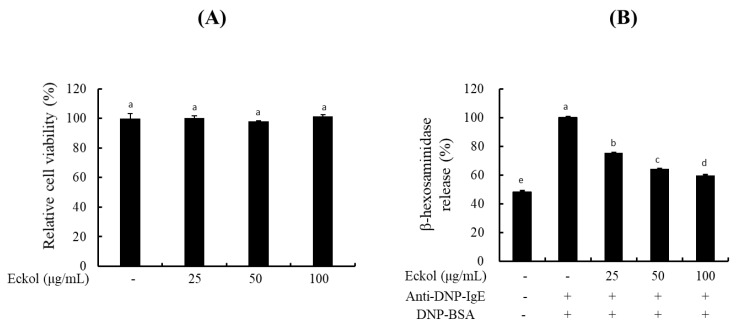
Effect of eckol on β-hexosaminidase release in IgE/BSA-stimulated BMCMC. (**A**) Cytotoxicity of eckol in BMCMC (**B**) β-hexosaminidase content in IgE/BSA-stimulated BMCMC. Data are expressed as the means ± SD (*n* = 3) of three individual experiments. Differences in mean value among groups were assessed by one-way analysis of variance followed by Duncan’s test using PASW statistics 21.0 software. A value of *p*-value < 0.05 was considered statistically significant.

**Figure 3 nutrients-12-01361-f003:**
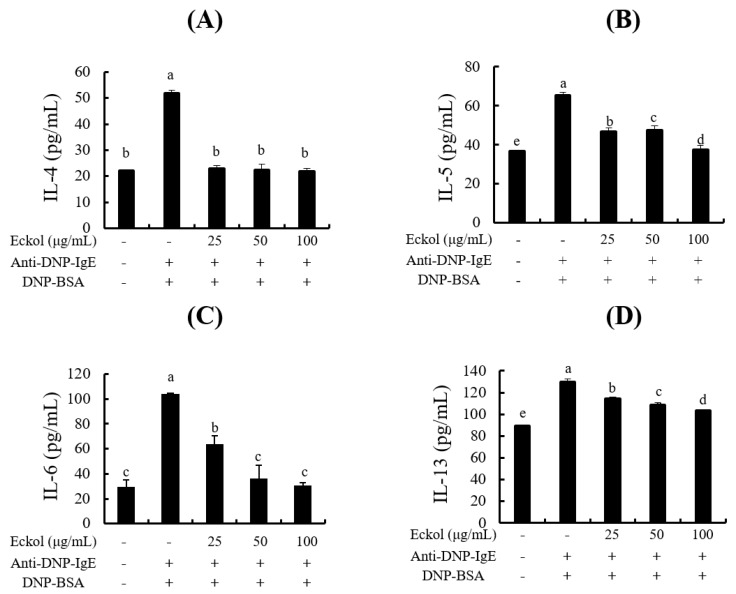
Effects of eckol on cytokine production in IgE/BSA-stimulated BMCMC. (**A**) IL-4, (**B**) IL-5, (**C**) IL-6 and (**D**) IL-13 production. Data are expressed as means ± SD (*n* = 3) of three individual experiments. Differences in mean value among groups were assessed by one-way analysis of variance followed by Duncan’s test using PASW statistics 21.0. A value of *p* < 0.05 was considered statistically significant.

**Figure 4 nutrients-12-01361-f004:**
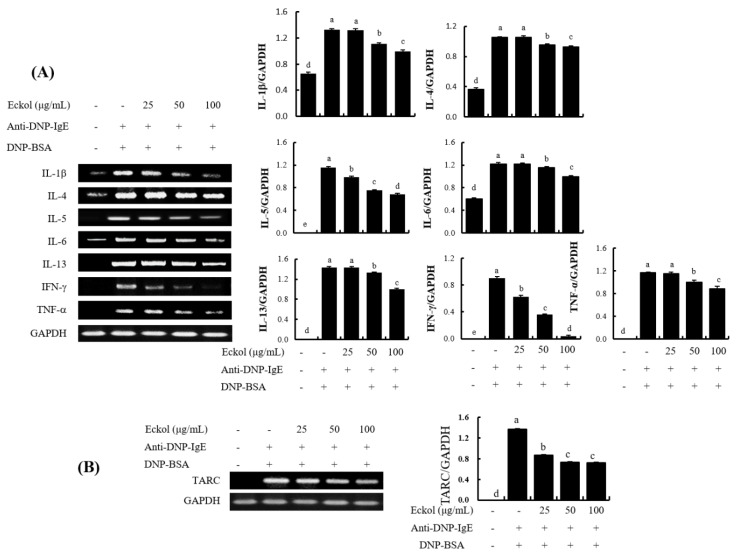
Effect of eckol on the gene expression of cytokines and chemokine in IgE/BSA-stimulated BMCMC. (**A**) mRNA expression of the cytokines and (**B**) the chemokine. Data are expressed as means ± SD (*n* = 3) of three individual experiments. Differences in mean values among group were assessed by one-way analysis of variance followed by Duncan’s test using PASW statistics 21.0. A value of *p* < 0.05 was considered statistically significant.

**Figure 5 nutrients-12-01361-f005:**
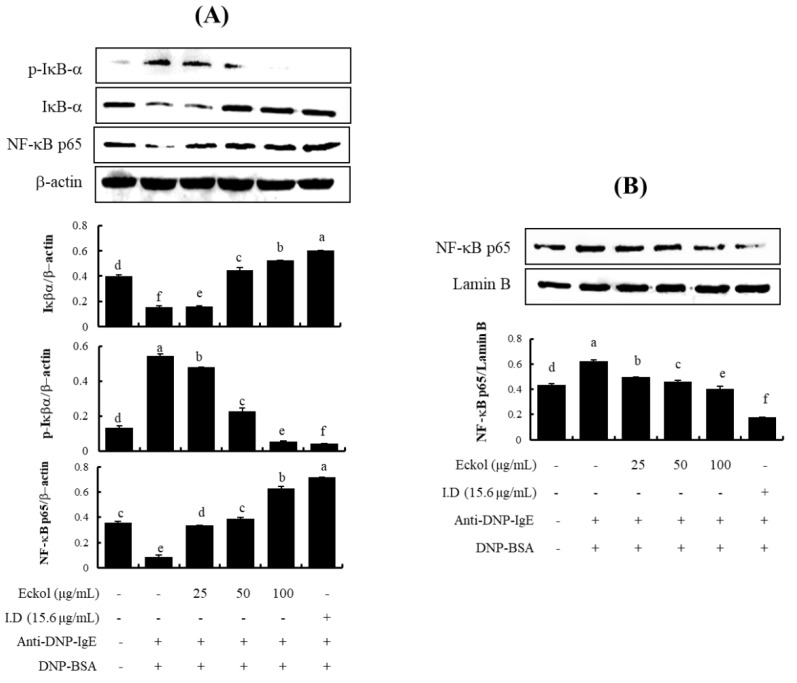
Effect of eckol on IκB-α and NF-κB activation in IgE/BSA-stimulated BMCMC. (**A**) IκB-α activity in the cytosol and (**B**) NF-κB activity in the nucleus. Data are expressed as means ± SD (*n* = 3) of three individual experiments. Differences in mean values among groups were assessed by one-way analysis of variance followed by Duncan’s test using PASW statistics 21.0. A value of *p* < 0.05 was considered statistically significant.

**Figure 6 nutrients-12-01361-f006:**
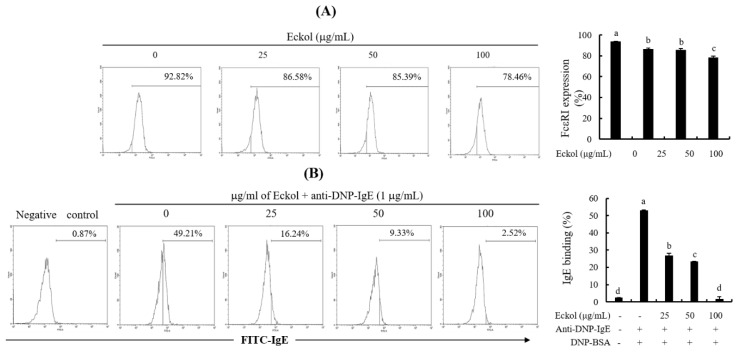
Effect of eckol on the cell surface expression of FcεRI and binding of IgE to FcεRI. (**A**) Cell surface FcεRI expression and (**B**) IgE antibody binding to FcεRI. Data are expressed as means ± SD (*n* = 3) of three individual experiments. Differences in mean values among groups were assessed by one-way analysis of variance followed by Duncan’s test using PASW statistics 21.0. A value of *p* < 0.05 was considered statistically significant.

**Figure 7 nutrients-12-01361-f007:**
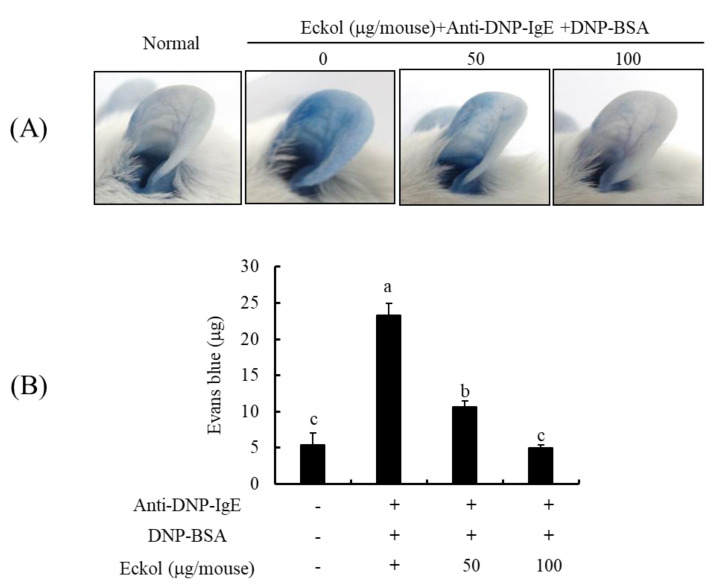
Effect of eckol on the IgE-mediated PCA reaction in mice. (**A**) Representative photographic images of ears, (**B**) Amount of extracted dye. Data are expressed as means ± SD (*n* = 3) of three individual experiments. Differences in mean values among groups were assessed by one-way analysis of variance followed by Duncan’s test using PASW statistics 21.0. A value of *p* < 0.05 was considered statistically significant.

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
