# Peer review of "Eckol from Ecklonia cava Suppresses Immunoglobulin E-mediated Mast Cell Activation and Passive Cutaneous Anaphylaxis in Mice"

_nutrients, 2020, doi:10.3390/nu12051361_

Round 1
Reviewer 1 Report
In their manuscript „Eckol from Ecklonia cava Suppresses IgE-mediated Mast Cell Activation and Passive Cutaneous Anaphylaxis in Mice“, Han et al. present data showing the suppressive nature of eckol on mast cell activation on the level of signal transduction as well as effector functions in vitro and in vivo. While it is, in general, very interesting to learn whether and how natural substances influence immune cell activation and hence potential disease progression, thorough analyses have to be performed to learn about effects, side effects, and mechanisms. In the present form, the reviewer cannot suggest publication, since several changes to the manuscript have to be incorporated and additional experiments to be performed.
Major points:
1) Figure 2A: they show that eckol is causing an enhanced reaction in a MTT assay and suggest that eckol is not exhibiting any cytotoxic effect. While this is correct, the data even show that eckol treatment leads to an increase in MTT positivity. This can be either enhanced cellular metabolic activation or proliferation. This has to be substantiated.
2) Figure 2B: the basic release from unstimulated cells is given as 100, the maximum release in response to antigen as approximately 210; what is the unit? percent? Either the release from the unstimulated cells is very high or the release from the stimulated cells quite low. The description here has to be much clearer.
Moreover, and that is true for all experiments, the n of independent experiments has to be mentioned in the Figure legends.
3) Eckol pretreatment completely suppresses IL-4 and IL-6 protein production independent of the eckol concentration used. This does not fit to the results shown in Figure 4, since here, the lowest concentration of eckol is not affecting Il4 and Il6 mRNA production at all and also the higher concentrations are nor as effective as suggested in Figure 3. Is eckol affecting the process of translation? This has to be explained.
4) Figure 5: in the figure legend, the stimulation length has to be mentioned. Moreover, in Figure 5B, the quality of cytosol-nucleus separation has to be proven. Is there lamin B in the cytosol fraction and is their tubulin only in the cytosol fraction and not in the nucleus fraction?
5) Figure 6: in A, the authors suggest that eckol pretreatment reduces the amount of FceRI on the cell surface as well as in B, eckol represses IgE binding to the FceRI. It has to explained why a reduction of FceRI expression from 92% to 78% (Figure 6A) translates into a reduction of IgE binding from 56% down to 1.6%! This does not make sense and is a very critical point for the description and understanding of the effect exerted by eckol. Is it possible that eckol in a mostly unselective fashion is binding to the cellular surface (glycocalyx) and thus prohibits binding of ligands and thus cellular reactions? This point of interaction of eckol with the mast cell is very important to be addressed.
Minor points:
1) In general, the manuscript should be read by a native speaker, since several sentences are incomplete or “out of grammar”.
2) line 24: “marie” has to be changed to “marine”
3) line 27: “in.” has to be removed
4) line 60/61: IL-3, -6, and -9 are mentioned as cytokines and growth factors. To mention them as cytokines is enough.
5) line 64: “crossing linking” has to be changed to “crosslinking”
6) chapter 2.4: the concentration of DNP-BSA has to be mentioned
7) line 113: after the degranulation reaction, the released beta-hexosaminidase is in the supernatant. Why would one wash the cells with 500 µL of buffer at this step?
8) line 136: why “twenty minutes later”? Is this important? please explain.
9) lines 177/178: change “stimulation of IgE/BSA” to “stimulation with IgE/BSA”
10) lines 201/202: p65 is right, p64 is wrong; this has to be changed.
Reviewer 2 Report
The authors well described and characterized the biological activity of eckol from the brown algae Ecklonia cava. I have found the article well write and very interesting.
I would suggest only some changes:
The authors report in par 3.1 the cell viability of BMCMCs in the presence of eckol,how do the author explain the increasing in cell viability (>100%)? I think this aspect should be clarified.
The authors should explain the anti-DNP-IgE treatment in the par 3.1 and not in par 3.4
The authors included a table containing the sequences of the primer in par 3.3
they should include these information in suppl mat or in method.
Figure 4: there are a lot of interesting information, but the picture of the agarose gels could be added as suppl mat
References: please check the style (in ref 12, 13, 14 and others the years of pubblication is not write in the same style of the number 1,2 or other)
Reviewer 3 Report
Eckol from Ecklonia cava Suppresses IgE-mediated Mast Cell Activation and Passive Cutaneous Anaphylaxis in Mice
Eui Jeong Han et al.
For authors:
Eui Jeong Han et al. analyzed the role of Eckol extracted from Ecklonia in type1 allergic reaction. Their data indicated that Eckol inhibits mast cell function such as degranulation and cytokine production. Furthermore, the authors showed that Eckol suppressed IgE/DNP-induced PCA reaction in vivo. Finally, the authors concluded Eckol extracted from Ecklonia exhibits anti-allergic activity in vitro and in vivo. The results proposed here are interesting. However, more in-depth analyses are necessary to confirm the authors’ conclusions. For example, evidence showing the molecular mechanism(s) underlying the suppression of allergic responses by Eckol is insufficient.
1, The authors are not proposing any molecular mechanism to explain the inhibitory effects of Eckol on allergic reactions. This study would be more informative if the mechanism by which Eckol exerts its anti-allergic effect. Why did Eckol suppress the cell surface expression of FceRI and binding of IgE antibody to FceRI?
2, The authors indicated that Eckol has no obvious cytotoxicity on mast cells. The data shown in Figure 2(A) indicates that 100 μg/ml of Eckol doubles the viability of mast cells in 24 hours. Such increase in the number of BMCMCs is usually not observed and is extremely surprising. Would you please provide detailed explanation of this data?
3, The authors checked the level of cytokine gene induction in Figure 4. However, some cytokine profiles are not consistent with the data shown in Figure 3. For example, in figure 3, they showed that Eckol can block IgE/DNP-mediated IL-6 production in mast cells. On the other hand, in Figure 4, there is no apparent suppression of IgE/DNP-induced IL-6. Please comment on this discrepancy.
4, They showed that Eckol inhibited IgE/DNP-induced down-regulation of IkB alpha and NF-kB translocation into the nucleus. I suggest that the authors check other signaling pathways such as Syk, LAT, and PLCg1.
5, In Figure 7, the authors showed that Eckol suppressed IgE/DNP-induced PCA reaction. They need to determine if compound inhibits vascular permeability.
6, Molecular size marker is necessary in Figure 5.
7, IL-4, IL-5, and IL-13 are Th2-type cytokines. They are not called inflammatory cytokines.
8, The manuscript requires thorough proofreading and revision to improve the quality of English. The authors need to address language errors that are seen throughout the manuscript.
For example:
- The first sentence in the introduction is grammatically incorrect and hard to understand.
- Line 44-45: “is contain in ~” should be “is contained in ~”
- Line 63: ”mast cells have been known to secretion~” should be ”mast cells have been known to secrete ~”.
Round 2
Reviewer 1 Report
In the revised version of their manuscript „Eckol from Ecklonia cava Suppresses IgE-mediated Mast Cell Activation and Passive Cutaneous Anaphylaxis in Mice“, Han et al. addressed a few of the reviewer´s points satisfyingly, HOWEVER, several of the major points were not really addressed and require further work. They will be added in red following the reviewer´s points from the first review (in black) and the authors´ response (in blue):
Major points:
1) Figure 2A: they show that eckol is causing an enhanced reaction in a MTT assay and suggest that eckol is not exhibiting any cytotoxic effect. While this is correct, the data even show that eckol treatment leads to an increase in MTT positivity. This can be either enhanced cellular metabolic activation or proliferation. This has to be substantiated.
→ Cell viability was retested and the results was replaced.
-> in the first version of their manuscript, the authors showed a dose-dependent increase in “survival” by eckol, and these changes were significant. In the revised version, the authors show no difference in the MTT assay. Though this might be the more likely result, it questions the reproducibility of the data presented by Han et al. At least it has to be explained to the reviewer AND the editor, why the data from the first submission were so different from the revised data!
2) Eckol pretreatment completely suppresses IL-4 and IL-6 protein production independent of the eckol concentration used (Fig. 3). This does not fit to the results shown in Figure 4, since here, the lowest concentration of eckol is not affecting Il4 and Il6 mRNA production at all and also the higher concentrations are nor as effective as suggested in Figure 3. Is eckol affecting the process of translation? This has to be explained.
→ Our results involving Figure 3 and 4, are to confirm the presence or absence of the inhibitory effects eckol of on cytokine production in DNP-IgE-induced allergic reactions, and it is difficult to do quantitative evaluation.
-> what does this mean: “it is difficult to do quantitative evaluation”? For Il5 and Ifn-g mRNA they show dose-dependent reduction by eckol; for these mRNAs, quantitative evaluation apparently was not difficult? Why is it difficult for Il4 and Il6 mRNA? They have to explain/discuss what their data can mean with respect to the effect of eckol on mast cell activation.
3) In Figure 5B, the quality of cytosol-nucleus separation has to be proven. Is there lamin B in the cytosol fraction and is their tubulin only in the cytosol fraction and not in the nucleus fraction?.
→ In general, it is used to show whether the cytosol and nuclear fraction are well done in western blot analysis. If separation was successful, only tubulin would be detected in cytosol fraction, and Lamin B would be detected in the nuclear fraction. Lamin B and tubulin can be used as protein loading controls for nuclei and cytoplasm, respectively.
Figure 5B was the result of NF-B expression in nucleus to assess the inhibitory effects of eckol on NF-B translocation from cytosol to nucleus. Therefore, the laoding control used lamin B.
-> lamin B should not only be used as a loading control, but more importantly as a control for the successful separation of nucleus and cytosol. For this reason, one has to prove that there is no contamination of the nuclear lysate with cytosol components. Therefore, the absence of tubulin has to be proven.
4) Figure 6: in A, the authors suggest that eckol pretreatment reduces the amount of FceRI on the cell surface as well as in B, eckol represses IgE binding to the FceRI. It has to be explained why a reduction of FceRI expression from 92% to 78% (Figure 6A) translates into a reduction of IgE binding from 56% down to 1.6%! This does not make sense and is a very critical point for the description and understanding of the effect exerted by eckol. Is it possible that eckol in a mostly unselective fashion is binding to the cellular surface (glycocalyx) and thus prohibits binding of ligands and thus cellular reactions? This point of interaction of eckol with the mast cell is very important to be addressed.
→ Binding of IgE antibody to FceRI was retested the results were replaced.
-> though the new data (Fig. 6B) are slightly different from the data of the first version of their manuscript, the quality of the results is the same: strong, almost 100% reduction of IgE binding by eckol pretreatment, however, only a weak attenuation of FceRI expression by eckol. The point made above by the reviewer has not been addressed at all and this point is critical for the understanding of the effect by eckol. The sentence added in lines 334/335 (mentioning what should e done) is not enough to address the problem with these data. At least the authors have to discuss the potential meaning of their data (even when disadvantageous).

Author Response
"Please see the attachment."

Reviewer 3 Report
The authors have nicely answered all my questions and the paper is substantially improved.
Author Response
Thank you very much for your kind review with more depth to our manuscript.
Round 3
Reviewer 1 Report
In the second revision of their manuscript „Eckol from Ecklonia cava Suppresses IgE-mediated Mast Cell Activation and Passive Cutaneous Anaphylaxis in Mice“, Han et al. again addressed a few of the reviewer´s points satisfyingly, however, some other points were not addressed and require additions to the manuscript. They will be added in bold red following the authors´ response to the second review:
Major points:
1) Eckol pretreatment completely suppresses IL-4 and IL-6 protein production independent of the eckol concentration used (Fig. 3). This does not fit to the results shown in Figure 4, since here, the lowest concentration of eckol is not affecting Il4 and Il6 mRNA production at all and also the higher concentrations are nor as effective as suggested in Figure 3. Is eckol affecting the process of translation? This has to be explained.
→ Our results involving Figure 3 and 4, are to confirm the presence or absence of the inhibitory effects eckol of on cytokine production in DNP-IgE-induced allergic reactions, and it is difficult to do quantitative evaluation.
-> what does this mean: “it is difficult to do quantitative evaluation”? For Il5 and Ifn-g mRNA they show dose-dependent reduction by eckol; for these mRNAs, quantitative evaluation apparently was not difficult? Why is it difficult for Il4 and Il6 mRNA? They have to explain/discuss what there data can mean with respect to the effect of eckol on mast cell activation.
->As shown in Fig. 4, we measured cytokine mRNA expression levels, including IL-4 and IL-6. However, there is a limit to visual confirm, so the densitometry was measured and presented together with the band. As a results, it was confirmed that the expression levels of IL-4 and IL-6 also decreased by Eckol in dose-dependently manner. Already, many papers have shown that IL-4 and IL-6 are associated with mast cell activation. Therefore, our results indicated that mast cell activation is related to IL-4 and IL-6.
-> concerning Fig. 4, the reviewer was not questioning at all the used technique (RT-PCR followed by densitometry). However, the dose-response analysis in Fig. 4 is not in agreement with the one presented in Fig. 3. 25 µg/ml eckol do not inhibit IL-4 and IL-6 mRNA expression at all and 50 and 100 µg/ml eckol only have a quite weak effect (Fig. 4). In Fig. 3, however, independent of the eckol concentration used, expression of IL-4 and IL-6 protein is suppressed to baseline. Since transcription is happening before translation, the reviewer wonders how these data do fit together, how this can be explained. Thus, the point of the reviewer from the first review was that the authors in this respect have to put some meaningful explanatory sentences into their manuscript (either in the results or the discussion section).
2) Figure 6: in A, the authors suggest that eckol pretreatment reduces the amount of FceRI on the cell surface as well as in B, eckol represses IgE binding to the FceRI. It has to be explained why a reduction of FceRI expression from 92% to 78% (Figure 6A) translates into a reduction of IgE binding from 56% down to 1.6%! This does not make sense and is a very critical point for the description and understanding of the effect exerted by eckol. Is it possible that eckol in a mostly unselective fashion is binding to the cellular surface (glycocalyx) and thus prohibits binding of ligands and thus cellular reactions? This point of interaction of eckol with the mast cell is very important to be addressed.
→ Binding of IgE antibody to FcRI was retested the results were replaced.
-> though the new data (Fig. 6B) are slightly different from the data of the first version of their manuscript, the quality of the results is the same: strong, almost 100% reduction of IgE binding by eckol pretreatment, however, only a weak attenuation of FceRI expression by eckol. The point made above by the reviewer has not been addressed at all and this point is critical for the understanding of the effect by eckol.
-> Based on the reviewer’s comments, two experiments were conducted to confirm the accuracy of the results. In both experiments, we confirmed that FceRI expression and IgE binding to the FceRI was suppressed by Eckol. Actually, these results showed a similar pattern to the mast cell degranulation inhibitory activity of Dieckol, a similar compound isolated from E.cava (Ahn, G., Amagai, Y., Matsuda, A., Kang, S. M., Lee, W., Jung, K., & Jeon, Y. J. (2015)). Dieckol, a phlorotannin of Ecklonia cava, suppresses IgE-mediated mast cell activation and passive cutaneous anaphylactic reaction. Experimental dermatology, 24(12), 968.). As far as we know, there is no reports of Eckol’s mechanism of action for the surface receptor FceRI. From our results, it can be confirmed that Eckol has an anti-allergic activity by effectively suppressing the FceRI expression and IgE binding to the FceRI, and more detailed research is needed.
-> the reviewer completely agrees that more detailed research is needed. Again, the reviewer does not question the data, but since, again, data do not fit together, some discussion of the results and what they could mean is necessary. This is why scientific publications do have a discussion section. Once again, as mentioned in the first review: the authors show that eckol is only weakly attenuating FceRI expression on the surface of the mast cells studied (Fig. 6A), however, it very strongly seems to suppress IgE binding to the receptor (Fig. 6B). Given that almost 80% of receptors are still present on the cells after using 100 µg/ml eckol (Fig. 6A), it is not logical, at first sight, why the binding of IgE is reduced to 2.52% in the same situation (Fig. 6B). The reviewer and also the potential future reader would like, at least, to read something about the thoughts, the authors have about this interesting, however, non-logical finding. It should also be discussed why their results with eckol are rather different to the results from Ahn et al. (2015. Experimental dermatology. 24:968), who studied the effects of Dieckol. In the study by Ahn et al., the dose-response effect of Dieckol on IgE binding (Fig. 1c) and on FceRI expression (Fig. 1d) did match very well, and hence was completely different to the results by Han et al.
